# REAL: Rectified Adversarial Sample via Max-Min Entropy for Test-Time Defense

## Abstract

Adversarial attacks expose the vulnerability of neural networks. But it is difficult for existing defense methods to defend against all attacks, which leads to the lack of generalization in adversarial robustness. Inspired by test-time adaptation which leverages model's prediction entropy to generalize naturally distributed samples during testing, we try to rationally utilize adversarial samples' entropy for sample rectification, and then achieve test-time defense. In this article, we investigate the entropy properties of adversarial samples and obtain two observations: 1) adversarial samples are often confidently misclassified despite having low prediction entropy and 2) samples with higher attack strength typically show lower prediction entropy. Therefore, we believe directly minimizing the entropy of adversarial samples is not reasonable and propose a two-stage self-adversarial rectification approach: Rectified Adversarial Sample via Max-Min Entropy for Test-Time Defense (REAL), consisting of a max-min entropy optimization scheme and an attack-aware weighting mechanism, which can be embedded in the existing models as a plugged-played block. Experiments on several datasets show that REAL can greatly improve the performance of existing sample rectification model.

## 1 Introduction

Adversarial attacks expose vulnerabilities of deep neural networks and arise the thinking about security issues of real application. To mitigate the dangers posed by adversarial attacks, adversarial defenses have emerged Szegedy et al. (2013); Kurakin et al. (2016); Madry et al. (2017); Zhang et al. (2019); Wang et al. (2019); Wong et al. (2020). However, most existing defense methods only perform well on known attacks but struggle to generalize unknown attacks, highlighting the generalization challenge in adversarial robustness Stutz et al. (2019); Tsipras et al. (2018); Su et al. (2018). Recent studies Wang et al. (2020); Zhang et al. (2022) focused on generalization in **natural robustness** inspire us. They follow a consensus that, for natural distribution samples, entropy is related to error rate. That is, the lower the entropy, the higher the prediction confidence and the lower the error rate. Then these works minimize model prediction entropy during testing to enhance the generalization in natural robustness.

This inspiration leads us to consider:

*Can we also leverage the entropy of adversarial samples during testing for test-time defense thus achieving the generalization in **adversarial robustness**?*

Before addressing this question, we need to know the entropy properties of adversarial samples by conducting statistical experiments as shown in Fig. 1. We present the statistical relationship between entropy and error rate of adversarial samples generated using PGD attack Madry et al. (2017) on CIFAR10 Krizhevsky et al. (2009) in Fig. 1a and observe adversarial samples exhibit an adversarial characteristic, i.e., they tend to be confidently misclassified despite having low prediction entropy, which is obviously different from those natural distribution samples. Therefore we believe direct using entropy minimization for test-time defense is not a viable approach. To make reasonable use of the entropy of adversarial samples, a natural question is *how to prevent adversarial samples from being misclassified*

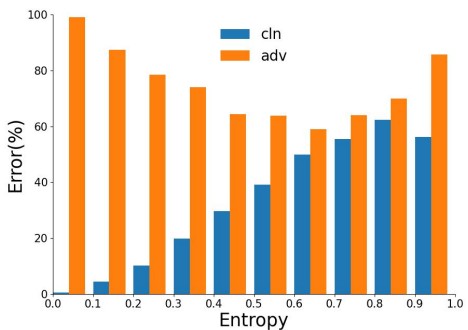 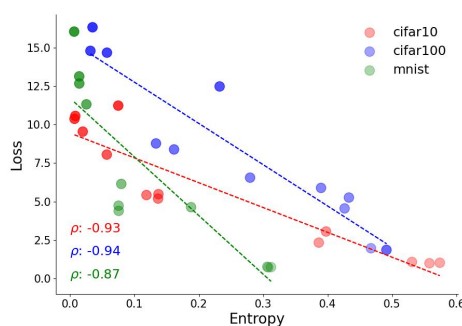

(a) observation 1: relationship between predicted entropy and error rates on adversarial samples generated by PGD attack and clean samples of CIFAR10.

(b) observation 2: greater attack strength causes more loss and lower entropy for adversarial samples across various attack methods (represented by different points) on three datasets.

Figure 1: The entropy properties of adversarial samples

*with high confidence?* A simple idea to answer this question is to transform these adversarial samples into ***mask samples*** by ***maximizing*** their entropy. Then we obtain mask samples that no longer exhibit strong adversarial characteristic. However, our ultimate goal is to obtain ***purified samples*** that can be classified correctly with high confidence. Hence, we further introduce to enhance the mask samples by ***minimizing*** entropy, thereby obtaining purified samples. Eventually, we can attain anticipated purified samples through the above two-stage max-min (i.e., maximizing entropy to obtain mask sample and minimizing entropy to obtain purified sample) entropy optimization scheme. Notably, we incorporate this max-min entropy mechanism into the existing models as a plugged-played block.

This max-min entropy mechanism is based on the assumption that adversarial samples are generally misclassified with high confidence, however this assumption may vary across different attackers. As shown in Fig. 1b, not all adversarial examples exhibit high confidence misclassification with low entropy. Therefore, we should employ the max-min entropy mechanism to varying degrees. Besides, we observe adversarial samples with higher classification loss and stronger attack strength tend to exhibit lower entropy. So we introduce an attack-aware weighting mechanism that exploits the attack strengths by assessing samples' predicted entropy, and combine it with max-min entropy optimization scheme. Thus an attack-aware max-min entropy optimization is formed.

Our contribution is elaborated as follows: (1) We explore the predicted entropy of adversarial samples and arrive at two observations. (2) We propose a two-stage self-adversarial rectification approach: Rectified Adversarial Sample via Max-Min Entropy for Test-Time Defense (REAL), consisting of a max-min entropy optimization scheme and an attack-aware weighting mechanism, and verify its effectiveness on several datasets.

## 2 RELATED WORK

**Information entropy** Information entropy measures event uncertainty in information theory. Small prediction entropy indicates higher prediction certainty Massey (1994), and it is often associated with lower prediction error rate. Wang et al. (2020); Zhang et al. (2022) explore this relationship on natural samples, and use this conclusion to achieve generalization in natural robustness during test-time adaptation through the minimization of information entropy. Wang et al. (2021) expands the application in the field of adversarial defense. However the previous research on prediction entropy do not consider the nature of adversarial samples. Therefore we explore the entropy properties of adversarial examples and propose a way to make rational use of the entropy.

**Adversarial attack** Since Szegedy et al. (2013) discovers the vulnerability of neural networks: adding some disturbances to images that cannot be distinguished by the human eye can lead to images being misclassified with high confidence by the network. And some works have also emerged to explore the vulnerability of neural networks Goodfellow et al. (2014); Athalye et al. (2018); Madry et al. (2017); Carlini & Wagner (2016); Moosavi-Dezfooli et al. (2016); Kurakin et al. (2016); Croce & Hein (2020); Dong et al. (2018); Xiao et al. (2018). Adversarial samples generated by different attack methods perform differently and this raises the challenge of generalization in adversarial defense.

**Adversarial rectification** Under various attacks, many defense methods have also emerged accordingly. Among them, adversarial training (AT) Kurakin et al. (2016); Madry et al. (2017); Zhang et al. (2019); Wang et al. (2019); Wong et al. (2020); Jia et al. (2022); Dong et al. (2023) is believed to be one of the most effective defense strategies. However, these method are considered to have poor generalization for unknown attackers and is computationally expensive for large-scale networks. Test-time defense strategy is proposed to effectively address this challenge, drawing inspiration from the domain adaptation (DA) field Boudiaf et al. (2022); Wang et al. (2020); He et al. (2020); Sun et al. (2019). These methods can be divided into the following two categories based on the criteria of where to adapt: i) Model adaptation, which aims to adapt the model during test time Wang et al. (2021); Chen et al. (2021); Kang et al. (2021); Gandelsman et al. (2022); Zhou et al. (2020). ii) Input adaptation, also known as adversarial rectification, which attempts to purify adversarial samples during inference. In this paper, we select the latter for implementation.

Shi et al. (2021); Mao et al. (2021); Hwang et al. (2023); Yang et al. (2022); Tsai et al. (2023) employ an auxiliary network Chen et al. (2020); He et al. (2020); Feng et al. (2019) for rectifying adversarial samples. Among them, Mao et al. (2021) adds auxiliary networks to the already robust model (trained via AT) for sample rectification which still relies on traditional adversarial training. Although Hwang et al. (2023) provides results on standard trained model, the improvement is not significant, and high accuracy can only be achieved when multiple defense strategies are combined. In addition, Wu et al. (2021) gives a method of rectifying samples without the aid of auxiliary network just by attacking all classes. We think its essence is similar to minimizing the predicted entropy of samples. But our observations suggest that it is inappropriate to simply minimize the entropy of a standard trained model and the experimental results of Wu et al. (2021) also indicate that for standard trained models, the improvement of sample rectification is not significant. In conclusion, previous works neither use the entropy of adversarial samples, nor exploit the entropy in a reasonable way in standard trained models. In this paper, we propose to delve into the prediction entropy of adversarial samples in test-time defense in a reasonable way.

## 3 METHOD

In this section, we introduce REAL consisting of a max-min entropy optimization scheme and an attack-aware weighting mechanism. The overall framework of our method is shown in Fig. 2.

### 3.1 PRELIMINARY

Adversarial rectification models can be summarized as the following paradigm: given a pretrained model with two branch structures, namely the main task and auxiliary task, where the main task represents the ultimate classification objective and the auxiliary task involves self-supervised objective such as data reconstruction Feng et al. (2019); Tsai et al. (2023); Yang et al. (2022), rotation prediction Gidaris et al. (2018) or label consistency He et al. (2020); Chen et al. (2020). These two branches share the encoding part defined as $E$. For a given input $x$, the encoder outputs $z = E(x, \theta_{enc})$. The classifier is represented as $C$ and outputs prediction $\hat{y} = C(z, \theta_{cls})$. The auxiliary task is represented as $A(z, \theta_{aux})$. During the training phase, the two branches conduct joint training with the training goal as Eq. 1. Here the $\mathcal{L}_{cls}$ is the cross entropy for classification, $\mathcal{L}_{aux}$ is the auxiliary self-supervised objective.

$$\min_{\theta} \mathcal{L}_{\text{train}} = \mathcal{L}_{cls}((C \circ E)(x), y, \theta_{enc}, \theta_{cls}) + \mathcal{L}_{\text{aux}}((A \circ E)(x), \theta_{enc}, \theta_{aux}) \qquad (1)$$

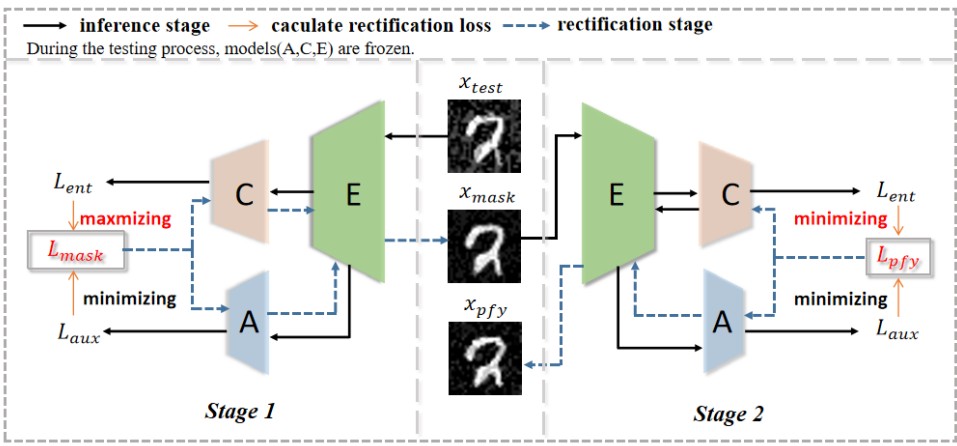

Figure 2: Overall framework of our method: REAL

In the testing phase, the sample is rectified based on the loss of auxiliary tasks $\mathcal{L}_{\text{aux}}(x)$. The calculation process is shown in Eq. 2, where $\epsilon_{\text{pfy}}$ is the budget of adversarial perturbation. The parameter $\delta$ refers to the perturbation added to sample and is computed through gradient optimization.

$$\min_{\delta} \mathcal{L}_{\text{aux}} \left( (A \circ E) (x + \delta) \right), \quad s.t. \|\delta\| \leq \epsilon_{\text{pfy}}, \ x + \delta \in [0, 1] \tag{2}$$

After obtaining the purified sample $x_{pfy} = x + \delta$, feed them into the model to obtain predictions as the final classification output $y_{pre} = (C \circ E)(x_{pfy})$.

### 3.2 MAX-MIN ENTROPY OPTIMIZATION SCHEME

In this paper, we delve into the properties of adversarial samples' prediction entropy. Observation 1 shows that adversarial samples are generally misclassified with low entropy and high confidence, which is obviously different from natural distribution samples. To answer this question of *how to use the entropy of adversarial samples reasonably*, we propose a max-min entropy optimization scheme that involves two steps. Firstly, we employ entropy maximization for adversarial sample $x_{adv}$ together with an auxiliary task loss, so as to disrupt the inherent property of adversarial sample and introduce a masking effect. We achieve this goal by optimizing the mask loss $\mathcal{L}_{\text{mask}}$ to obtain a mask sample $x_{mask} = x_{adv} + \delta$ as shown in Eq. 3, where $\beta_{\text{max}}$ is a trade-off parameter between the two losses and is associated with attack strength, as discussed in section 3.3.

$$\mathcal{L}_{\text{mask}} = \mathcal{L}_{\text{aux}} - \beta_{\text{max}} \cdot \mathcal{L}_{\text{ent}} \tag{3a}$$

$$\min_{\delta} \mathcal{L}_{\text{mask}} \left( x_{adv} + \delta; A, C, E \right), \quad s.t. \|\delta\| \leq \epsilon_{\text{pfy}}, \ x_{adv} + \delta \in [0, 1] \tag{3b}$$

After the first stage, we believe that the adversarial characteristics of samples have been diminished. Subsequently, in the second stage, we further apply entropy minimization to enlighten the mask sample $x_{mask}$ to obtain purified sample $x_{pfy} = x_{mask} + \delta$ which can be correctly classified with high confidence by optimizing the purified loss $\mathcal{L}_{\text{pfy}}$, as shown in Eq. 4, where $\beta_{\text{min}}$ is a trade-off parameter between the two losses as discussed in Sec. 3.3.

$$\mathcal{L}_{\text{pfy}} = \mathcal{L}_{\text{aux}} + \beta_{\text{min}} \cdot \mathcal{L}_{\text{ent}} \tag{4a}$$

$$\min_{\delta} \mathcal{L}_{\text{pfy}} \left( x_{\text{mask}} + \delta; A, C, E \right), \quad s.t. \ \|\delta\| \leq \epsilon_{\text{pfy}}, \ x_{\text{mask}} + \delta \in [0, 1] \tag{4b}$$

The above two-stage max-min entropy optimization scheme can be termed as "adversarial rectification". During the process of adversarial rectification, the predicted values of the adversarial samples engage in a game of deterministic and uncertain predictions, ultimately producing purified samples that meet rectification cutoff conditions, to be discussed in Sec. 3.4.

### 3.3 ATTACK-AWARE WEIGHTING MECHANISM

Regarding observation 2 as shown in Fig. 1b, it indicates the variations in the predicted entropy of adversarial samples generated by different attack methods and adversarial samples with higher attack strength exhibit lower predicted entropy values. Inspired by this, we further propose an attack-aware weighting mechanism that takes the attack strength into consideration by introducing a dynamic rectification parameter $\beta$. First, we assess the attack strength of an adversarial sample by calculating its predicted entropy value $\mathcal{V}_{ent}$, which is normalized by dividing $log(N)$, N is the number of categories. For adversarial samples with lower attack strength, i.e. higher $\mathcal{V}_{ent}$, thus it requires less extent of entropy maximization. Therefore we design $\beta_{\max}$ to dynamically adjust the entropy maximization degree, as shown in Eq. 5, where $\alpha$ represents a trade-off hyperparameter.

$$\beta_{\max} = \alpha \cdot (1 - \mathcal{V}_{\mathrm{ent}})^2, \quad \mathcal{V}_{\mathrm{ent}} \in [0, 1] \tag{5}$$

On the contrary, after the first stage of adversarial characteristics destruction, we obtain mask samples and need to inspire them by entropy minimization. And for these mask samples with higher predicted entropy values, we need to apply greater extent of entropy minimization. We employ it by designing $\beta_{\min}$ as described in Eq. 6.

$$\beta_{\min} = \alpha \cdot (\mathcal{V}_{\mathrm{ent}})^2, \quad \mathcal{V}_{\mathrm{ent}} \in [0, 1] \tag{6}$$

### 3.4 OVERALL ALGORITHM: MULTI-STEP OPTIMIZATION

We provide a comprehensive algorithm structure by combining the above two components. Algorithm 1 summarizes our procedure. In our rectification algorithm, to avoid significant accuracy decrease on clean samples, we first introduce a clean/adversarial sample detection step. Taking inspiration from Tsai et al. (2023), we investigate the utilization of auxiliary loss for detection and observe that adding entropy loss can enhance sample detection. Fig. 3 illustrates the distributions of auxiliary loss and entropy loss for clean and adversarial samples. From the figure, we can observe that setting both auxiliary loss thresholds $aux^*$ and entropy loss thresholds $ent^*$ can serve as a means of detection, effectively distinguishing between clean and adversarial samples. The specific implementation is: for a given input $x$, if it simultaneously satisfies conditions of $\mathcal{L}_{aux}(x) < aux^*$ and $\mathcal{L}_{ent}(x) < ent^*$, we classify it as a clean sample and output it without further rectification. Otherwise, we perform REAL for sample rectification.

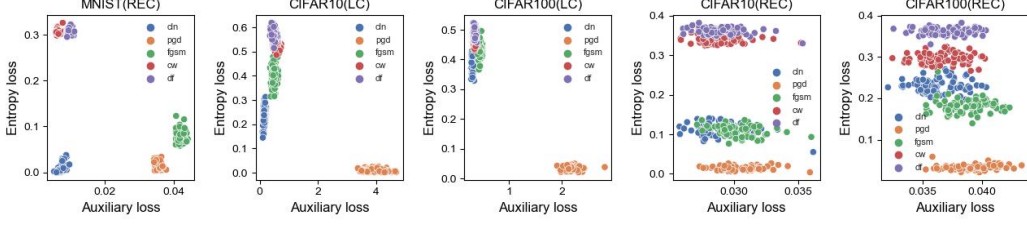

Figure 3: The joint distributions of auxiliary loss and entropy loss. Clean samples (cln in blue) show different distribution from adversarial samples under various attacks.

However self-adversarial rectification process causes sample predictions to oscillate between deterministic and uncertain states, therefore the rectification becomes more challenging. Thus one round optimization may not be sufficient. To overcome this, inspired by multi-step attack Madry et al. (2017) we adopt a multi-step optimization by increasing the number of self-adversarial rectification rounds. To fix the number of optimization rounds, we have further developed a heuristic selection strategy (hss) of rectification rounds number by outputting purified samples if they satisfy rectification cutoff condition which are determined by Eq. 7.

$$\underbrace{[\mathcal{L}_{\mathrm{aux}}(x_{\mathrm{pfy}}) < aux^* \text{ and } \mathcal{L}_{\mathrm{ent}}(x_{\mathrm{pfy}}) < ent^*]}_{\text{subcondition1}} or \underbrace{[\mathcal{L}_{\mathrm{aux}}(x_{\mathrm{pfy}}) < aux^* \text{ and } LR]}_{\text{subcondition2}} \tag{7}$$

---

**Algorithm 1** Rectified Adversarial Sample via Max-Min Entropy for Test-Time Defense

---

**Input:** $x$: Test sample, $aux^*$: Auxiliary loss detection threshold, $ent^*$: Information entropy detection threshold, $T$: the number of rectification steps in one stage, $\gamma$: Purification step size, $R$: the maximum number of rectification rounds.
**Output:** $x_{\text{pfy}}$: Purified sample.
  1: **if** $\mathcal{L}_{aux}(x) < aux^*$ and $\mathcal{L}_{\text{ent}}(x) < ent^*$ **then**
  2:     Return $x_{\text{pfy}} = x$
  3: **else**
  4:     Initialize $\delta = 0$, $x_{\text{pfy}} = x$, $rounds = 1$
  5:     **repeat**
  6:        **for** $t$ in range($2T$) **do**
  7:            Compute normalized predication entropy of sample: $\mathcal{V}_{\text{ent}}(x_{\text{pfy}})$
  8:            **if** $t < T$ **then**
  9:                Compute $\beta_{\max}$ according to Eq. 5;
 10:                Compute $\mathcal{L}_{\text{mask}}$ according to Eq. 3;
 11:                Perform a single step gradient descent based on $\mathcal{L}_{\text{mask}}$:
 12:                $\delta = \delta - \gamma \cdot \text{sign}\left(\nabla_x \mathcal{L}_{\text{mask}}\right)$.
 13:            **else**
 14:                Compute $\beta_{\min}$ according to Eq. 6;
 15:                Compute $\mathcal{L}_{\text{pfy}}$ according to Eq. 4;
 16:                Perform a single step gradient descent based on $\mathcal{L}_{\text{pfy}}$:
 17:                $\delta = \delta - \gamma \cdot \text{sign}\left(\nabla_x \mathcal{L}_{\text{pfy}}\right)$.
 18:            **end if**
 19:            $x_{\text{pfy}} = x + \delta$
 20:        **end for**
 21:        $rounds \mathrel{+}= 1$
 22:     **until** $x_{\text{pfy}}$ meets condition in Eq. 7 or $rounds > R$
 23: **end if**

---

It involves two subconditions. The first is relatively strict, requiring that both auxiliary loss $\mathcal{L}_{\text{aux}}(x_{\text{pfy}})$ and information entropy of purified sample $\mathcal{L}_{\text{ent}}(x_{\text{pfy}})$ are below a certain threshold. This allows for screening out clean samples that are not detected. The second is less stringent, requiring that auxiliary loss of purified sample $\mathcal{L}_{\text{aux}}(x_{\text{pfy}})$ falls below threshold, while also verifying label reversal (LR), which refers to the inconsistency between label predictions of purified sample and original sample, LR: $(C \circ E)(x_{pfy}) \neq (C \circ E)(x)$.

## 4 EXPERIMENTS

### 4.1 EXPERIMENTAL SETTINGS

**The selection of self-supervised tasks** Our method can be combined as a plugged-played block with existing models that exploits self-supervised tasks for sample rectification. In specific experiments, our method is combined with data reconstruction and label consistency tasks. The reconstruction task Feng et al. (2019); Tsai et al. (2023); Yang et al. (2022) employs an autoencoder to align the decoder's output with the input image, helping the model understand the image's internal distribution. The label consistency task He et al. (2020); Chen et al. (2020), commonly used in self-supervised learning methods, utilizes contrastive learning to ensure predictions of augmented images align with those of the original images.

**The selection of datasets and backbone** We validate our method on MNIST Lecun & Bottou (1998), and CIFAR-10/CIFAR-100 Krizhevsky et al. (2009). We only utilize the reconstruction task as an auxiliary task for MNIST due to the potential label changes caused by certain image augmentation techniques like rotation. For CIFAR10/100, we assess both the reconstruction and label consistency tasks. We use different backbone architectures for each dataset: FCN and CNN architectures for MNIST, and ResNet18 He et al.

(2016), WideResNet-28-10 Zagoruyko & Komodakis (2016) architectures for CIFAR10/100, respectively. For fair comparison, we adopt the same parameters as Shi et al. (2021).

**The selection of attack methods** We conduct experiments on several common attacks, including FGSM, PGD, CW, and DeepFool(DF). At the same time, further experimental analysis is conducted on adaptive attacks (i.e. attacking with all defense strategies known), ultimately demonstrating the effectiveness of our method. For MNIST, perturbation budget are set $\epsilon=0.3$, and PGD takes 40 steps with step size of 0.01. CW, DF are set with the same parameter as Shi et al. (2021). For CIFAR10/100, perturbation budget are set $\epsilon=8/255$, and PGD takes 20 steps with step size of 2/255. CW, DF are set with the same parameter as Shi et al. (2021).

**The selection of defense method parameters** Initially, we use auxiliary loss and entropy loss thresholds to detect clean and adversarial samples, as depicted in Fig. 3. However, we observe that this method is ineffective in distinguishing clean and adversarial samples for reconstruction task on CIFAR10/100. The complexity of reconstructing these datasets makes it challenging to accurately fit clean samples during training, resulting in a mixed distribution of reconstruction losses, as Fig. 3 shows. Therefore, we choose not to employ threshold detection for data reconstruction task. Instead, for label consistency task, we approximate the detection threshold based on statistical data obtained from clean samples. Our strategy employs a two-stage rectification approach. To reduce computational overhead, we set the number of iterations $T$ in each stage to 3 for three datasets. Besides we set an iteration step $\gamma$ of 0.1 for MNIST and 4/255 for CIFAR10/100. During the rectification phase, we set the trade-off parameter $\alpha$ to 0.25 and the maximum number of rectification rounds $R$ to 5.

## 4.2 MAIN RESULTS

We compare adversarial training, represented by AT (FGSM) Goodfellow et al. (2014) and AT (PGD) Madry et al. (2017), and sample rectification method SOAP Shi et al. (2021) with our proposed two-stage rectification method. The auxiliary in Table 1,2,3 means adding an auxiliary network for joint training. The brackets indicate the different auxiliary tasks used, where (lc) and (recs) represent label consistency auxiliary and data reconstruction task respectively. We reproduce SOAP Shi et al. (2021) using same parameters in this paper and paste the results of original paper in brackets. The optimal **values** are highlighted in both bold and underlined, while the suboptimal **values** are shown in bold.

Table 1: Adversarial robust accuracy on the MNIST test set.

| Method | FCN(Fully Connected Neural network) | | | | | CNN(Convolutional Neural Network) | | | | |
| --- | --- | --- | --- | --- | --- | --- | --- | --- | --- | --- |
| | Natural | FGSM | PGD | CW | DF | Natural | FGSM | PGD | CW | DF |
| NO defense | **98.10** | 16.87 | 0.49 | 0.01 | 1.40 | **99.15** | 1.49 | 0.00 | 0.00 | 0.00 |
| AT (FGSM) | 79.76 | **80.57** | 2.95 | 6.22 | 17.24 | 98.78 | **99.50** | 33.70 | 0.02 | 6.16 |
| AT (PGD) | 76.82 | 60.70 | 57.07 | 31.68 | 13.82 | 98.97 | **96.38** | **93.22** | 90.31 | 75.55 |
| auxiliary (rec) | 97.90 | 34.67 | 1.11 | 0.17 | 2.12 | **99.06** | 69.95 | 30.24 | 0.14 | 6.34 |
| SOAP (rec) | **97.47(97.56)** | 70.46(66.85) | **64.29(61.88)** | **95.10(86.81)** | **93.82(87.02)** | 99.06(98.94) | 85.04(87.78) | 76.89(84.92) | 87.49(74.61) | **86.10(81.27)** |
| SOAP+ours (rec) | 97.39 | **95.85** | **96.82** | **98.28** | **97.60** | 98.85 | 93.97 | **92.38** | **98.05** | **96.42** |

Table 2: Adversarial robust accuracy on the CIFAR10 test set.

| Method | resnet18 | | | | | widresnet28-10 | | | | |
| --- | --- | --- | --- | --- | --- | --- | --- | --- | --- | --- |
| | Natural | FGSM | PGD | CW | DF | Natural | FGSM | PGD | CW | DF |
| NO defense | **90.54** | 15.42 | 0.00 | 0.00 | 6.26 | **95.13** | 14.82 | 0.00 | 0.00 | 3.28 |
| AT (FGSM) | 72.73 | 44.16 | 37.40 | 2.69 | 24.58 | 72.20 | 91.63 | 0.01 | 0.00 | 14.41 |
| AT (PGD) | 74.23 | 47.43 | 42.11 | 3.14 | 25.84 | 85.92 | 51.58 | 41.50 | 2.06 | 24.08 |
| auxiliary (rec) | 83.24 | 12.88 | 1.59 | 0.00 | 10.31 | 85.87 | 23.46 | 7.30 | 0.06 | 10.94 |
| SOAP (rec) | 78.10 | 24.29 | 17.29 | 66.50 | 65.97 | 76.67 | 32.38 | 23.46 | 62.07 | 64.61 |
| SOAP+ours (rec) | 67.77 | 37.11 | 31.52 | 67.42 | 65.52 | 72.48 | 41.6 | 35.32 | 69.47 | 67.49 |
| auxiliary (lc) | **86.42** | 22.04 | 0.15 | 0.00 | 8.62 | **93.69** | 57.24 | 3.08 | 0.02 | 43.43 |
| SOAP (lc) | 83.85(84.07) | **52.45(51.02)** | 48.82(51.42) | 82.83(73.95) | 81.88(74.79) | 91.11(91.89) | 64.56(64.83) | 56.39(53.58) | 82.82(80.33) | 59.73(60.56) |
| SOAP+ours (lc) | 78.82 | **58.29** | **62.43** | **85.8** | **82.01** | 91.02 | **65.78** | **58.55** | 83.54 | 59.87 |

For MNIST, our experimental results are significantly improved compared with other methods. And under some attack methods such as CW and DF on both model structures, better results than adversarial training have been achieved. For most attacks, our method

Table 3: Adversarial robust accuracy on the CIFAR-100 test set.

| Method | resnet18 | | | | | widresnet28-10 | | | | |
|---|---|---|---|---|---|---|---|---|---|---|
| | Natural | FGSM | PGD | CW | DF | Natural | FGSM | PGD | CW | DF |
| NO defense | **65.56** | 3.81 | 0.01 | 0.00 | 12.30 | **78.16** | 13.76 | 0.06 | 0.01 | 9.05 |
| AT (FGSM) | 44.35 | 20.30 | 17.41 | 4.23 | 18.15 | 46.45 | **88.24** | 0.15 | 0.00 | 13.40 |
| AT (PGD) | 42.15 | 21.92 | 20.04 | 3.57 | 17.90 | 62.71 | 28.15 | **21.34** | 0.65 | 16.57 |
| auxiliary (rec) | 52.36 | 4.12 | 0.70 | 0.00 | 13.06 | 63.44 | 15.78 | 6.05 | 0.14 | 11.87 |
| SOAP (rec) | 52.46 | 9.34 | 6.74 | 40.36 | 41.19 | 53.59 | 21.36 | 17.77 | 42.73 | 42.87 |
| SOAP+ours (rec) | 37.14 | 14.51 | 10.80 | 41.51 | 39.40 | 48.46 | 23.78 | 20.04 | 46.16 | 45.02 |
| auxiliary (lc) | **58.67** | 7.99 | 0.04 | 0.00 | 12.74 | **74.28** | 18.72 | 0.64 | 0.00 | 10.02 |
| SOAP (lc) | 56.49(52.91) | **26.85(22.93)** | 25.39(27.55) | **55.26(50.26)** | **55.49(50.57)** | 63.91(61.01) | 31.57(31.4) | 37.17(37.53) | **57.01(56.09)** | **54.02(53.79)** |
| SOAP+ours (lc) | 44.27 | 30.49 | **35.54** | 55.02 | 51.6 | 56.57 | **31.69** | **39.03** | 53.97 | 50.5 |

can achieve the optimal defense effect, which also shows our method can alleviate non-generalization in adversarial robustness. For CIFAR10/100, using label consistency task as auxiliary network can achieve the optimal effect in our method and the overall results are superior to traditional adversarial training methods. Compared with SOAP (lc) Shi et al. (2021), our method can further improve the accuracy under most attacks, by 4%/6% under FGSM attack. In the data reconstruction task, although the optimal accuracy is not achieved, it can still be observed that the effect is significantly improved after adding our method.

For all three datasets, our method demonstrates significant enhancement under PGD and FGSM attacks compared with other attacks (e.g., CW and DF). We also provide an explanation for this phenomenon. As shown in Fig. 3, under PGD and FGSM attacks, adversarial samples exhibit high-confidence misclassification and significantly different from the entropy distribution of clean samples. Consequently, upon integrating the max-min entropy self-adversarial rectification strategy, notable improvements in performance can be observed. In contrast, CW and DF operate on different principles and are designed to introduce subtle perturbations leading to misclassifications, resulting in a smaller difference between the entropy distributions of adversarial and clean samples. In this regard, our method can still achieve some improvement effects, for example, it can improve by nearly 3% on CIFAR10(resnet) as shown in Table 2. However, we also observe on CIFAR100, our method is slightly inferior to SOAP at defending against CW and DF. The reason may be due to the poor detection results on these attack methods. As shown in Fig. 3, for CIFAR100, the distribution of clean samples and adversarial samples generated by CW and DF are mixed together. Therefore, it difficult to detect them by setting thresholds and most adversarial samples are incorrectly identified as clean samples, leading to insufficient optimization. Hence, in future research, the detection method for purified samples can be optimized for better sample rectification.

### 4.3 ABLATION ANALYSIS

In defense phase, we propose a max-min entropy optimization scheme and an attack-aware weighting mechanism. In addition, we propose a heuristic selection strategy (hss) of rectification rounds. We verify the effectiveness of these three parts on CIFAR10. Results are shown in Table 4, from which we observe that by employing max-min entropy optimization (+max-min) can improve the rectification effect of samples to a certain extent. After adding the heuristic selection strategy (+hss), the rectification accuracy can be significantly improved. In addition, we have seen that after adding attack-aware weighting mechanism $(+\beta)$, rectification effect can be further improved.

Table 4: Ablation results on CIFAR10 to verify the effectiveness of max-min entropy optimization scheme, attack-aware weighting mechanism and heuristic selection strategy respectively.

| Method | resnet(rec) | | | | | | resnet(lc) | | | | | |
|---|---|---|---|---|---|---|---|---|---|---|---|---|
| | Natural | FGSM | PGD | CW | DF | AVG | Natural | FGSM | PGD | CW | DF | AVG |
| NO defense | **83.24** | 12.88 | 1.59 | 0.00 | 10.31 | 21.60 | **86.42** | 22.04 | 0.15 | 0.00 | 8.62 | 23.45 |
| SOAP | 78.10 | 24.29 | 17.29 | 66.50 | 65.97 | 50.43 | 83.85 | 52.45 | 43.82 | 82.83 | 81.88 | 68.97 |
| SOAP+max-min | 77.10 | 22.75 | 12.58 | 66.75 | 66.35 | 49.11 | 84.16 | 55.09 | 54.44 | 81.27 | 80.36 | 71.06 |
| SOAP+max-min+hss | 67.20 | 36.89 | **31.84** | 67.33 | 65.35 | 53.72 | 77.79 | **58.38** | 61.20 | 84.40 | 81.20 | 72.59 |
| SOAP+max-min+hss+$\beta$ | 67.77 | **37.11** | 31.52 | **67.42** | **65.52** | **53.87** | 78.82 | 58.29 | **62.43** | 85.8 | **82.01** | **73.47** |

### 4.4 RESULTS ON DEFENSE AWARE ATTACK

In order to further verify the reliability of our method, we discuss a powerful adaptive attack method, i.e., defense aware attack (DAA), that is, attacking the model with prior knowledge of all defense strategies. In defense stage, we reduce auxiliary loss and rectify samples through max-min entropy optimization based on adversarial properties that prediction entropy of adversarial samples is usually small. Therefore, during defense aware attack, in addition to maximizing classification loss, we also minimize auxiliary loss to decrease its role in defense and reversely modify the property by maximizing entropy of adversarial samples, thus designing a defense aware attack, as shown in Eq. 8, where $\sigma$ is the trade-off parameter that can be adjusted to simulate different attack strength.

$$\mathcal{L}_{\mathrm{DAA}} = \mathcal{L}_{\mathrm{cls}} - \mathcal{L}_{\mathrm{aux}} + \sigma * \mathcal{L}_{\mathrm{ent}} \tag{8}$$

By maximizing the DAA loss $\mathcal{L}_{\mathrm{daa}}$, we can get adversarial samples by this adaptive attack. Experimental results as shown in Fig. 4, confirm that our method remains effective in defending against this attack, even when all defense strategies are well-known.

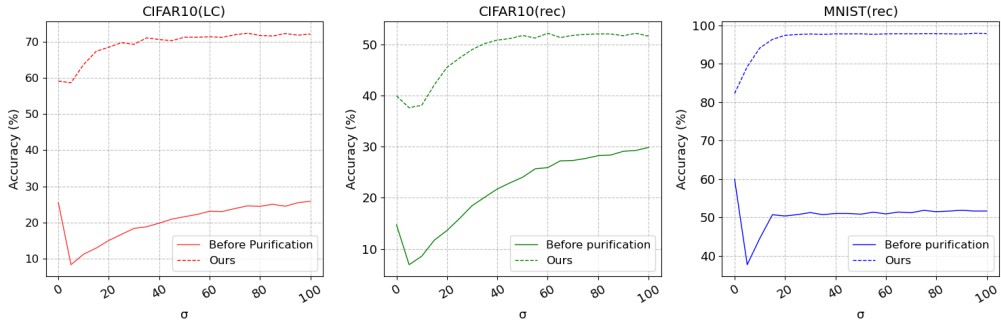

Figure 4: Results for defending defense aware attack. Plots are classification accuracy before (solid line) and after (dashed line) rectification using REAL.

When $\sigma$ is smaller than a certain range, the attack effect is stronger (i.e., accuracy shows a downward trend). But our method is still effective for defending this stronger attack. And when the weight $\sigma$ increases to a certain level, the attack effect will show a downward trend, which shows that the attacker cannot benefit from the prior knowledge of defense strategy. Overall, our method can provide a good defense against this adaptive attack because we introduce an entropy game process in defense process, making it difficult to reverse attack our method even if the defense strategy is known.

## 5 CONCLUSION

In this article, we investigate the entropy properties of adversarial samples and discover two important observations. Then we explore a reasonable way to utilize entropy by introducing a max-min entropy optimization scheme and an attack-aware weighting mechanism. Through experiments on three datasets, we successfully demonstrate that our proposed method has significant effectiveness in improving the generalization in adversarial robustness.

Limitations: Different choices of auxiliary tasks can significantly impact the detection performance. For instance, data reconstruction task cannot be used effectively for sample detection on CIFAR10/100 datasets. Moreover, the selection of the detection threshold has a substantial influence on the final outcome. Consequently, in our method, the selection and adjustment of detection thresholds are critical. Furthermore, the final rectification accuracy is also influenced by the choice of auxiliary tasks. When there is good correlation between the auxiliary tasks and the main task, it can further enhance the rectification effect. However, selecting appropriate auxiliary tasks can be challenging and requires further exploration in future work.

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

# 6 Appendix

## 6.1 Results on the TinyImageNet dataset

To showcase additional results on the TinyImageNet-200 dataset, we conduct partial experiments. TinyImageNet-200 is designed for image classification tasks. It consists of 120,000 training images, 10,000 validation images, and 10,000 test images, with each image sized at 64x64 pixels. The analysis was performed on two architectures, namely ResNet18 and WideResNet28-10. The experimental results are presented in Table **??** below:

Table 5: Adversarial robust accuracy on the TinyImageNet test set.

| Method | resnet18 | | | | | widresnet28-10 | | | | |
|---|---|---|---|---|---|---|---|---|---|---|
| | Natural | FGSM | PGD | CW | DF | Natural | FGSM | PGD | CW | DF |
| NO defense | **51.01** | 2.00 | 1.58 | 0.00 | 11.85 | **65.89** | 8.96 | 0.26 | 0.00 | 9.66 |
| AT (FGSM) | 29.00 | **13.45** | 12.29 | 8.42 | 18.19 | 50.78 | **24.00** | 19.97 | 10.25 | 24.65 |
| AT (PGD) | 28.49 | **13.70** | **12.98** | 8.42 | 17.60 | 49.45 | **22.99** | 19.18 | 8.95 | 26.87 |
| auxiliary (lc) | **44.45** | 1.64 | 0.00 | 0.00 | 12.34 | **64.32** | 15.70 | 1.12 | 0.00 | 9.66 |
| SOAP (lc) | 41.97 | 7.97 | 5.69 | **40.29** | **40.65** | 47.23 | 21.2 | **25.09** | **44.12** | **42.88** |
| SOAP+ours (lc) | 30.90 | 11.14 | **13.51** | **39.65** | **36.24** | 50.80 | 22.55 | **29.45** | **44.45** | **42.20** |

## 6.2 The necessary of mask loss

To illustrate the necessity of the mask stage, we will demonstrate it through entropy distribution plots and visualizations. First, we plot the entropy distribution on three datasets, CIFAR10, CIFAR100, and TinyImageNet, under PGD attacks for two different architectures, ResNet18 and WideResNet28-10, with the auxiliary task of label consistency. As shown in Figure 567, it can be observed that after the mask stage, samples are no longer misclassified with high confidence, and the overall entropy distribution moves closer to that of clean samples. Following the purify stage, the entropy distribution becomes closer to that of clean samples.

For a more intuitive understanding of the effects of the mask and purify stages, we provide some visual results in Figure 8. We sample images from MNIST dataset with FCN architecture for the rec task and from CIFAR-10 dataset with ResNet18 architecture for label consistency auxiliary task. After the mask stage, it can be observed that the confidence of predictions for the incorrect class gradually decreases. In the attention maps, focus on the incorrect class is disrupted, demonstrating that the mask stage plays a role in masking adversarial samples. Subsequently, after the purify stage, high-confidence predictions for the correct class gradually recover, and the attention maps progressively approach those of clean samples.

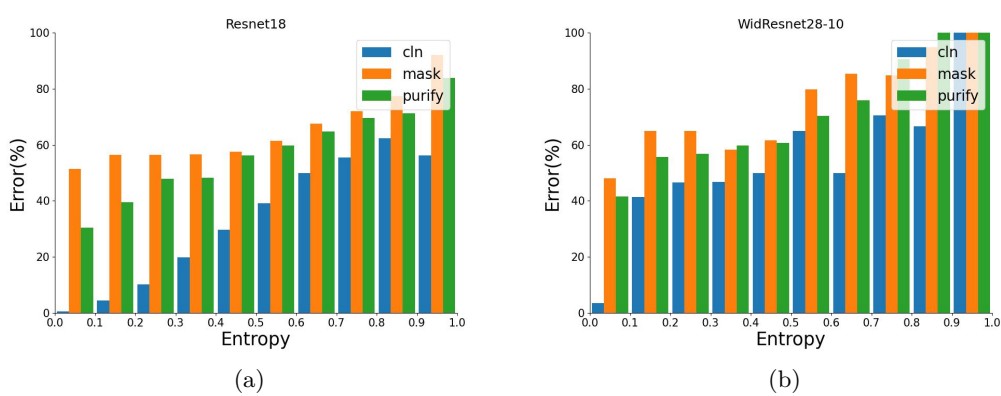

Figure 5: The entropy distribution plot during sample rectification stage on CIFAR10

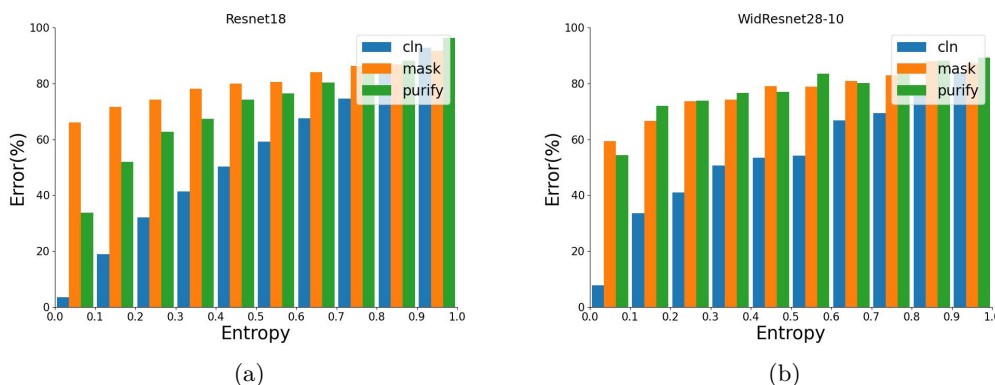

Figure 6: The entropy distribution plot during sample rectification stage on CIFAR100

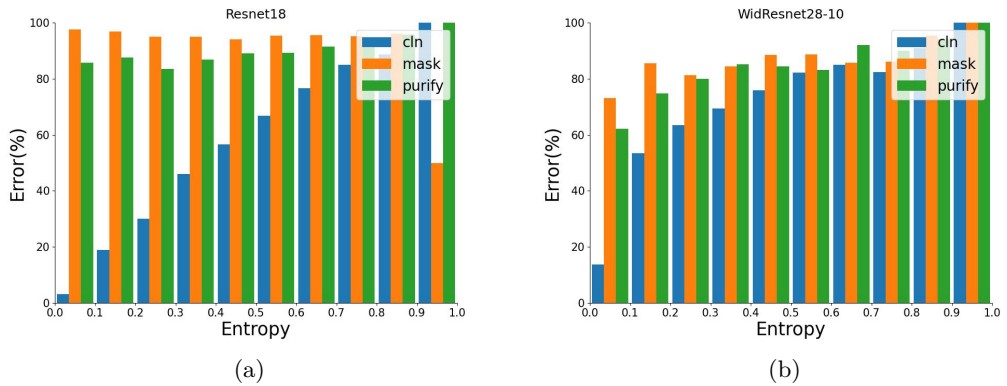

Figure 7: The entropy distribution plot during sample rectification stage on TinyImageNet

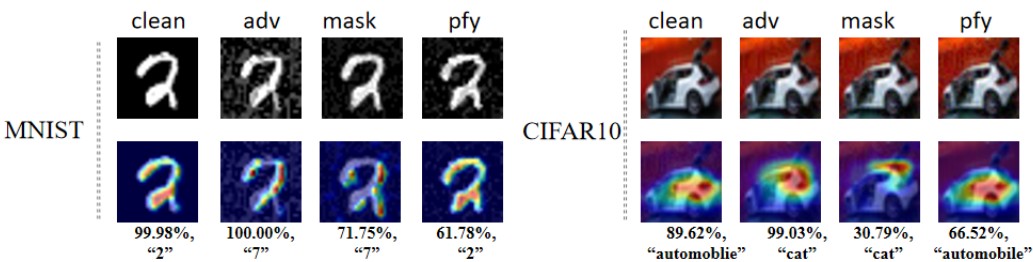

Figure 8: The visual results of the mask and purify stage.

