# OpenReview forum: "REAL: Rectified Adversarial Sample via Max-Min Entropy for Test-Time Defense"
_ICLR.cc/2024/Conference — Submitted to ICLR 2024_

### Official Review · Reviewer_7jdE · 2023-10-28

**Soundness:** 3 good
**Presentation:** 4 excellent
**Contribution:** 4 excellent
**Rating:** 8
**Confidence:** 4

**Summary:**

Improving adversarial robustness against adversarial attacks is an important but challenging task. This paper presents a critical problem that generalizing to numerous unseen adversarial attacks is difficult but paid less attention in the community, and proposes a new concept, i.e. generalizable robustness. Inspired by test-time adaptation, this paper proposes a new test-time defense methodology in robustness and overcomes the non-reasonable prediction entropy assumption in defense, and designs a two-stage rectification approach, i.e, REAL, through a max-min entropy optimization with attack-aware weighting mechanism. This submission brings some new perspectives that will promote adversarial robustness against unknown attacks. Experiments on benchmark datasets show the proposed REAL greatly improves the robustness of previous sample rectification models.

**Strengths:**

1. The idea of REAL is interesting, rational, and novel. First, it is valuable to explore new generalizable adversarial defense approaches against unseen attacks. Second, considering that adversarial perturbations are diverse and unseen in real applications, it is rational to establish a test-time training paradigm for removing perturbations. Third, since such a paradigm is still seldom studied in this field, the proposed test-time adversarial sample rectification approach is novel in multiple aspects, which may produce a high impact in related fields.
2. The proposed max-min entropy optimization strategy is new. The authors clearly claim that in conventional test-time adaptation for natural image classification, the entropy loss is commonly used for training unlabeled target data, and successfully reveal that this is not appropriate for adversarial samples as shown in Fig.1. Therefore, the authors propose a natural and novel idea to maximize the entropy of adversarial samples (stage 1) instead of minimization. For the final objective of accurate recognition, stage 2 is formulated for minimizing the entropy of rectified adversarial samples. The two-stage rectification paradigm achieves test-time defense on the fly.
3. Another merit of this paper is the proposed attack-aware weighting mechanism, which is simple but useful. The intuition behind this is clear because each sample should be treated unequally due to the differences in their attacking power. This paper contributes a simple metric of attack strengths by assessing samples' prediction entropy.
4. Experiments on benchmark datasets fully prove the superiority of the proposed REAL method, by plugging and playing in the classical SOAP model.

**Weaknesses:**

1. Since the auxiliary task is leveraged during test-time optimization, the authors could discuss some choices of different auxiliary tasks, although one may not decide which one is better without empirical observation.
2. Since this work aims to improve generalization, a more complete setting toward attack generalization can be implemented in the future and produce a higher impact.
3. It is better to clarify which module is frozen and noted in Figure 2.

**Questions:**

1. Is the code of the proposed REAL approach available? This is also important to improve the impact on the open-source community.
2. In Fig.2, are the C and E frozen during test-time max-min optimization?
3. In Eq. 3b and 4b, what is "S" ? which is not defined. I guess this is a typo error, which may be "C".

---

> ### Author Response · Authors · 2023-11-17
> **To Reviewer 7jdE**
>
> We thank you for your reviews and will respond to your questions and concerns one by one.
>
> **Q1: The code**
>
> **A1:** As our paper is currently under review and considering the review process and requirements, we have not provided a code link at the moment. In the future, we plan to organize our code and make it publicly available on GitHub.
>
> **Q2: The model in test time**
>
> **A2:** In practice, our approach involves updating samples during the test time process using pre-defined fixed models. Therefore, during the test time stage, models A, C, and E have all been frozen, and only the samples are rectified using max-min entropy optimization to achieve defense objectives. The update process of sample is same as the process of generating adversarial samples, which is to update sample at the pixel level by backpropagating the gradient of a loss. And for a better understanding, we modify Fig.2 in the article and annotate the model in the testing stage.
>
> **Q3: The Eq 3b, 4b**
>
> **A3:** Thank you for bringing up this question. There is a mistake in the formula editing in Eq.3b and Eq.4b. In fact, here S should represent A and C. To understand these formulas, $L_{\text {mask}}$ involves two parts of the loss, namely $L_{\text {aux}}$ and$L_{\text {ent}}$. The computation of $L_{\text {aux}}$ involves networks A and E, while the computation of $L_{\text {ent}}$ involves networks C and E. Therefore, it should be represented as $L_{\text {mask}}(x_{adv}+\delta; A, C, E)$ in this context, and similarly in Eq4. We have corrected this error in the original text.

---

### Official Review · Reviewer_nJ3p · 2023-11-01

**Soundness:** 2 fair
**Presentation:** 1 poor
**Contribution:** 1 poor
**Rating:** 3
**Confidence:** 4

**Summary:**

This paper proposes a test-time adversarial defense that uses a combination of auxiliary task loss thresholds, entropy thresholds, and two sets of self-adversarial rectification rounds. The method is applied to adversarial defense on MNIST, CIFAR-10, and CIFAR-100, and it is observed that the method can provide robustness to an unsecured classifier.

**Strengths:**

* The method investigates an ambitious and relevant task of test-time adversarial defense.
* A wide variety of ideas from many different defenses are integrated into a novel method.

**Weaknesses:**

* The presentation of the method and the motivation of the difference aspects is somewhat difficult to follow.
* The primary weakness is that the method appears to be built upon a broken defense, namely the SOAP model. The work [a] reports breaking the SOAP defense using BPDA. I expect that a similar attack could be used against this method. Although this work does present an adaptive attack, from what I can tell the attack does not differentiate through the purification. BPDA provides an efficient way to do this approximately. Re-evaluation of this defense using the methodology in [a] is essential, especially given that this methodology has broken the SOAP defense this work is based on.
* The method does not compare with recent diffusion-based purification defenses such as [b], which generally obtain stronger results than those reported in this work.

[a] https://arxiv.org/pdf/2202.13711.pdf (ICML 2022)

[b] https://arxiv.org/pdf/2205.07460.pdf (ICML 2022)

**Questions:**

Can the authors re-evaluate their defense using the adaptive attack used against SOAP in [a]?

---

> ### Author Response · Authors · 2023-11-17
> **To Reviewer nJ3p**
>
> We thank you for your reviews and will respond to your questions and concerns one by one.
>
> **Q1:** The presentation of  method and motivation of the difference.
>
> **A1:** For a clearer understanding, we will briefly summarize the presentation of our motivation and method here.
>
> ### 1. A summary of our approach
> The core of our approach lies in exploring the properties of adversarial samples, specifically by investigating a way to utilize the entropy of adversarial sample properties—the max-min entropy optimization mechanism. Additionally, we propose an attack-aware adaptive weighting scheme. We present our proposed method as a plugin that can be embedded into the auxiliary task learning framework (ATL) to achieve defense objectives by rectifying samples during test time. Additionally, due to the introduction of a max-min entropy optimization mechanism, the predicted values of samples oscillate between confident and uncertain predictions during the optimization process, making the sample optimization more challenging. Considering the potential issue of insufficient optimization and to avoid over-optimization, we propose a combination algorithm to assist in achieving more stable sample rectification.
>
> ### 2. The differences from previous methods.
> In the past, there have been methods attempting sample rectification through the ATL framework, but they predominantly focus on the design and innovation of the auxiliary network within the ATL framework and do not adequately leverage the classification information inherent in the main task. In this paper, we explore the utilization of adversarial sample entropy, enabling the extraction of classification information from the main task during the test stage, even in the absence of labels. This, in turn, assists in better sample rectification.

---

> ### Author Response · Authors · 2023-11-17
> **To Reviewer nJ3p**
>
> **Q2:** The BPDA+APGD attack effect on our method
>
> **A2:** Here, we first introduce the implementation principle of BPDA+APGD and then report the test results of BPDA+APGD.
> ### 1. A brief explanation for BPDA+APGD attack
>
> We analyze the implementation principle of BPDA+APGD in work [a]. The specific implementation of this attack can be briefly understood as follows: during the attack process, instead of using only the gradient of the model with respect to the final purified image as the update direction (sign), the average gradient of the intermediate iterations generated during the rectification process is taken. Intuitively, this directs all intermediate images towards misclassification, making the attack more effective. And in the attack, this iteration is run 1000 times.
>
> ### 2. Implement and test BPDA+APGD on our method
>
> We attempt to apply the same BPDA+APGD strategy to our rectification method following the settings in [a] and obtain the results as shown in the following table.
>
> **Overview of main experimental contents:**
>
> We conduct attacks on both the rectification process of SOAP and our rectification method, obtaining two sets of adversarial samples denoted as `SOAP_BPDA` and `ours_BPDA`. The difference between the two sets of adversarial samples lies in attacking different rectification processes. Subsequently, we test both sets of adversarial samples using SOAP defense and our rectification method. Following the settings in [a], tests are conducted on CIFAR-10 dataset using ResNet18 architecture for label consistency auxiliary task.
>
> #Table1 BPDA+APGD on CIFAR10
>
> |            |  SOAP_BPDA |  ours_BPDA |
> |:----------:|:----------:|:----------:|
> | NO_defense |    4.00%   |    8.00%   |
> |    SOAP    |    3.60%   | **22.00%** |
> |    ours    | **23.40%** |   18.70%   |
>
> **Performance on SOAP_BPDA adversarial examples:**
>
> From Table 1, it can be observed that `SOAP_BPDA` adversarial samples achieve a 4% success rate against the base model without rectification. However, after SOAP rectification defense, the test results do not improve; instead, there is a slight decrease to 3.6%. This is consistent with the findings reported in [a] and indicates that for methods utilizing auxiliary tasks for sample rectification through an Auxiliary Task Learning (ATL) framework for defense, BPDA+APGD could be a potent attack strategy.
>
> Next, we employ our rectification method to defend against the `SOAP_BPDA` adversarial samples, resulting in a test success rate of 31.6%, demonstrating improvement. This result can be considered as a manifestation of the effectiveness of our method against black-box attacks, hence the observed enhancement is a reasonable outcome.
>
> **Performance on Ours_BPDA adversarial examples:**
>
> Then, to explore a more powerful attack, we introduce a second type of adversarial samples, `ours_BPDA`, where the attack directly targets our rectification process. From Table 1, it can be observed that the `ours_BPDA` adversarial samples, without any defense and at the same iteration count of 1000, exhibit an improvement in test accuracy on the base model from 4% to 8%. This shows that it is more difficult for an attacker to completely attack our rectification process.
>
> Additionally, when we apply rectification defense to `ours_BPDA` adversarial samples using our method, we still achieve a considerable improvement, obtaining a test accuracy of 18.7%. This implies that even if attackers have complete knowledge of our rectification process and specifically target it during the attack, our method can still provide a certain level of defense.
>
> **Cause Analysis:**
>
> Analyzing the underlying reasons, firstly, our defense process incorporates a max-min entropy optimization mechanism, making it harder for attackers to identify the gradient direction in such attacks, thus rendering the attack more difficult. Secondly, our defense method introduces an attack-aware weighting mechanism that dynamically adjusts weights with input variations, achieving real-time adaptability during test time and making it difficult to fully trace our rectification process.
>
> **Summary:**
>
> Furthermore, we conduct experiments on CIFAR-100 dataset, and it can be observed that our method can resist the BPDA+APGD attack designed in [a] to a certain extent.
>
> #Table2 BPDA+APGD on CIFAR100
>
> |            | SOAP_BPDA  | ours_BPDA |
> |------------|------------|-----------|
> | NO_defense | 1.70%      | 2.10%     |
> | SOAP       | 1.30%      | **9.20%** |
> | ours       | **10.00%** | 5.00%     |
>
> \[a\] https://arxiv.org/pdf/2202.13711.pdf (ICML2022)

---

> > ### Author Response · Authors · 2023-11-17
> > **To Reviewer nJ3p**
> >
> > **Q3:** Comparison with Diffusion-Based Purification Method
> >
> > **A3:** In this section, we will explain the differences between our method and the diffusion purification model and the prospects for our method.
> >
> > ### 1. The difference between our method and diffusion purification model
> >
> > The diffusion purification model has achieved optimal results in image rectification strategies, relying on large models and substantial computational resources. *DiffPure* exploits the mathematical principles in the diffusion process. Given an adversarial example, it undergoes a forward diffusion process with a small amount of noise, followed by a reverse generation process to restore a clean image. It explores a purification framework that utilizes diffusion models to refine images.
> >
> > In contrast, our work primarily explores the utilization of adversarial sample entropy, proposing a max-min entropy optimization scheme. We integrate this scheme into sample rectification methods using an auxiliary learning framework (ATL), thereby enhancing the defense capabilities of such methods. In our experiments, we incorporate the method into the *SOAP* framework, which is lighter than *DiffPure*. Additionally, compared to *DiffPure*, our training process is simpler and does not necessitate the involvement of adversarial samples. We solely learn an auxiliary network joint model using clean samples.
> >
> > ### 2. The prospects for our method
> >
> > Considering the comparison with the diffusion model, it is challenging for existing lightweight rectification networks to achieve results comparable to the diffusion model. Embedding the entropy mechanism into a more refined ATL architecture may be necessary to achieve comparable outcomes. It might be essential to embed the entropy mechanism into a more refined ATL architecture to bridge this gap. Regarding this point, in our experiments, we observe that by embedding the entropy mechanism into the ATL framework, better rectification effects can be achieved when the distributions learned by the auxiliary task and the main task are more consistent and closely connected. This observation is demonstrated by the comparison between the label consistency task and the reconstruction task in our experiments. The joint learning problem between the auxiliary network and the main task in auxiliary task learning (ATL) is also gradually being overcome, as focused in [b]. We believe that such methods can help us achieve better sample rectification, but further exploration is needed.
> >
> > \[b\] (https://arxiv.org/abs/2301.12618) (NIPS2023)

---

> > > ### Comment · Reviewer_nJ3p · 2023-11-22
> > > **New results might not be strong enough**
> > >
> > > Thanks so much to the authors for their efforts to investigate stronger attacks against their model. If my understanding is correct, using a strong adaptive attack significantly lowers the reported adversarial robustness, for example from ~60% to ~20% on CIFAR-10. Is that an accurate description of the new tables? If so, I am not sure that the results of this defense are close enough to established methods like DiffPure, even if the method is much faster.

---

> > > > ### Author Response · Authors · 2023-11-22
> > > > **To  Reviewer nJ3p**
> > > >
> > > > We have reevaluated your concern, which mainly focuses on the effectiveness of the adaptive attack against our method. Additionally, there are concerns about whether it can achieve the same effect as DiffPure under adaptive attacks.
> > > >
> > > > We will answer from two aspects. Firstly, starting from the design principles of adaptive attacks, we will explain why our method does not show numerically competitive results. Secondly, starting from our comparative experiments, we will provide an understanding and analysis of our method in a fairer setting.
> > > >
> > > > ### The Principle of Adaptive Attack
> > > >
> > > > We first need to clarify that the specific working principles of adaptive attacks on different models are also different. In fact, the BPDA+APGD adaptive attack we measured on our method [a] is already an improved adaptive attack. It takes into account the intermediate process of correction and 1000 iterations in the hyperparameter settings, making the attack very powerful. However, in general methods and strategies for testing adaptive attacks, only one backward differential fitting process is often considered, similar to the adaptive attack approach reported in DiffPure [b], instead of 1000 consecutive backward differentials set in our testing criteria. We believe this is one of the reasons why our method is not competitive under this adaptive attack, and we believe that making direct comparisons with DiffPure under the current different adaptive attack methods would be unfair.
> > > > In addition, it is still worth considering in the future to further explore the robustness of DiffPure and study whether there are improved adaptive attacks that can further weaken its defense capabilities.
> > > >
> > > > ### Our Comparative Experiments
> > > >
> > > > We compare our method with SOAP under this adaptive attack and consider it a fair setting. Experimental results indicate that our method maintains certain defense capabilities under the strong adaptive attacks explored in [a]. It is worth noting that, with the same number of iterations for backward differentiation, our method shows a trend of being more challenging to attack. On CIFAR-10, the attack success rate increases from 4% to 8%. Furthermore, under this strong adaptive attack, our correction strategy remains effective, with a defense rate of 18.7%, compared to 3.6% for SOAP. This suggests that our proposed entropy optimization strategy is resistant to being completely traceable by such a powerful adaptive attacker, resulting in stronger defense capabilities.
> > > >
> > > > However, we also acknowledge that our method still has some gap under such a strong adaptive attack, and the defense effect of 18.7% is still unsatisfactory. There is still significant room for improvement, which is worth exploring in the future.
> > > >
> > > > \[a\]https://arxiv.org/pdf/2202.13711.pdf
> > > >
> > > > \[b\]https://arxiv.org/pdf/2205.07460.pdf

---

### Official Review · Reviewer_f1E9 · 2023-11-08

**Soundness:** 3 good
**Presentation:** 3 good
**Contribution:** 2 fair
**Rating:** 3
**Confidence:** 3

**Summary:**

This paper investigates the properties of prediction entropy in adversarial samples and presents several strategies for adversarial defense. Specifically, it introduces a max-min entropy optimization scheme and an attack-aware weighting mechanism. The results demonstrate that these approaches are compatible with existing models and exhibit strong performance.

**Strengths:**

- The starting point is novel and the proposed attack-aware weighting mechanism is technically sound.
- The paper is generally well-written and easy to follow. The authors have illustrated their settings and motivations using bullet points to provide a clear understanding of their objectives.

**Weaknesses:**

- The limitation of selecting the detection threshold and auxiliary tasks is crucial yet challenging. Besides, in numerous real-world attack scenarios, the specific attack methods are often unknown.
- The motivation behind employing a max-min optimization scheme is unclear. Why is a mask loss necessary in this context?
- Additionally, the experiments conducted seem insufficient, and it would be beneficial to observe more results obtained on ImageNet.
- In the text, $L_{ent}$ and $L_{cls}$ are not consistent.
- In the preliminary, you'd better provide more introduction and explain about the $\delta$.

**Questions:**

- If entropy is related to error rate, what about mutual information or signal-to-noise ratio (SNR)? Do they have a similar effect?
- The effectiveness of the max-min optimization scheme lacks convincing evidence. Could you please include a comparison of the robust accuracy of $x_{mask}$ to support your claim further?

---

> ### Author Response · Authors · 2023-11-17
> **To Reviewer f1E9**
>
> We thank you for your reviews and will respond to your questions and concerns one by one.
>
> **Q1:  the limitations of auxiliary tasks and threshold selection**
>
> **A1:**
> We reiterate the focus of our approach: our approach focuses on exploring the utilization of adversarial sample entropy and combines it into the process of using the auxiliary network for sample rectification.
>
> Firstly, for the auxiliary task designed in the combined sample rectification process, we have maintained an open choice. Different auxiliary tasks will exhibit varying effects due to their inherent correlation with the main task. The better the correlation between the auxiliary task and the features learned by the main task, the more favorable the final rectification results will be. Our focus was not on exploring the auxiliary tasks themselves but rather on attempting to enhance the rectification effects on different auxiliary tasks by incorporating the entropy optimization mechanism we proposed.
>
> Secondly, we need to clarify the relationship between the selection of the threshold and the auxiliary task. The primary purpose of this threshold is to distinguish between clean and adversarial samples, preventing over-rectification and avoiding the issues of insufficient optimization. In the specific operation, we drew inspiration from [a], utilizing auxiliary task loss for detection to separate clean samples from adversarial samples. The reason for this approach is based on an assumption that auxiliary tasks will exhibit different characteristics on clean/adversarial samples. In our case, we further leverage this dissimilarity for detection. Furthermore, this assumption is indeed the underlying premise and foundation of all methods that utilize semi-supervised auxiliary tasks for sample purification. We believe that the principle behind this detection is reasonable.
>
> However, in practical applications, some auxiliary tasks may not strictly adhere to this ideal assumption, leading to the issues mentioned in the paper where some auxiliary tasks do not perform well in detection. This may be attributed to the joint learning between auxiliary and main tasks, leading to negative transfer, as mentioned in [b]. In other words, there is a conflicting optimization between the joint feature distributions learned by auxiliary and main tasks. Indeed, our experimental results confirm this observation. For instance, as shown in Table 2 and 3, when jointly training with auxiliary tasks, the classification accuracy on clean samples for the main task decreases. Moreover, the drop is more pronounced in the reconstruction task, indicating a more significant negative transfer effect from the reconstruction task to the main task due to lower correlation.
>
> The mismatch between the auxiliary task and the main task results in a significant performance gap for samples in both tasks, thereby affecting our detection phase. However, we believe this issue is addressable. Alleviating conflicts between the main task and auxiliary task, ensuring their learning consistency, and employing learning strategies like [b] during the training phase can assist in improving our detection performance.
>
> Currently, in the original ATL framework, we can still gain benefits by incorporating some auxiliary tasks with less apparent effectiveness, such as the reconstruction task used in the paper.
>
> Finally, we would like to provide a detailed explanation of our threshold selection strategy. In the actual selection, we follow the prior statistical data of natural clean samples. The underlying reason is that, during the training phase, our method exclusively focuses on clean samples, thus assuming that the statistical characteristics of the auxiliary task differ between clean and adversarial samples. In this case, when facing various complex unknown attacks in real-world scenarios, it does not affect the choice of the threshold.
>
> In summary, our main focus is on exploring the utilization of adversarial sample entropy and integrating it into methods relying on auxiliary tasks for sample rectification. Some auxiliary tasks may have limitations due to their inability to well distinguish between clean and adversarial samples. We believe that this issue can be mitigated through certain ATL methods, and currently, even when the auxiliary network is limited, we can still attain some improvements.
>
> [a] [Test-time defense against adversarial attacks: Detection and reconstruction of adversarial examples via masked autoencoder](https://arxiv.org/abs/2303.12848)
> [b] [ForkMerge: Mitigating Negative Transfer in Auxiliary-Task Learning](https://arxiv.org/abs/2301.12618)

---

> > ### Author Response · Authors · 2023-11-17
> > **To Reviewer f1E9**
> >
> > **Q2: the necessary of mask loss**
> >
> > **A2:** Here, we restate the motivation behind our approach. Initially, recognizing the distinct entropy properties of adversarial samples compared to clean samples, we propose the use of a mask loss to correct the entropy properties of samples aiming to achieve a masking effect on the samples and disrupt their adversarial characteristics of being misclassified with high confidence. Furthermore, we achieve an enlightening effect on masked samples through a pfy loss. This enables the samples to be correctly classified with high confidence, similar to clean samples.
> >
> > To better showcase the masking effect during the intermediate process, we demonstrate the robust performance after the mask stage and the robust performance after the purify stage, as shown in Table 1. (Table 1 is conducted on CIFAR-10, utilizing the ResNet18 and focuses on the task of label consistency.)
> >
> > #Table1 robustness results of mask samples and purify samples with maximum rounds of 5
> >
> > |        |            | FGSM   | PGD    | CW     | DF     |
> > |--------|:----------:|--------|--------|--------|--------|
> > |        | NO defense | 22.04% | 0.15%  | 0.00%  | 8.62%  |
> > |        | SOAP       | 52.45% | 48.82% | 82.83% | 81.33% |
> > | round1 | mask       | 48.60% | 43.58% | 81.85% | 81.34% |
> > |        | pfy        | 53.58% | 54.41% | 80.99% | 79.26% |
> > | round2 | mask       | 57.04% | 59.37% | 84.09% | 81.07% |
> > |        | pfy        | 57.16% | 59.51% | 84.71% | 81.80% |
> > | round3 | mask       | 58.16% | 62.11% | 85.53% | 82.03% |
> > |        | pfy        | 58.06% | 61.79% | 85.66% | 82.07% |
> > | round4 | mask       | 58.27% | 62.46% | 85.69% | 81.91% |
> > |        | pfy        | 58.30% | 62.29% | 85.66% | 82.00% |
> > | round5 | mask       | 58.36% | 62.44% | 85.83% | 82.03% |
> > |        | pfy        | 58.29% | 62.43% | 85.80% | 82.01% |
> >
> > In the table, we present the test results for mask and purify samples after the mask and purify stages. In our rectification algorithm, considering the possibility of insufficient optimization, we set the rectification rounds to be automatically determined, with a maximum of 5 rounds. Therefore, in Table 1, we provide robustness test results for the mask and purify stages after 5 rounds of rectification. It can be observed that, in the early round of the rectification phase, the robustness of the samples significantly improves after the mask stage, and the adversarial properties of the adversarial samples are partially disrupted. Subsequently, the purify stage enlightens the samples, further reducing the error rate. Due to the consideration of insufficient sample optimization, threshold filtering is performed after each stage, continuing the mask-purify stage updates for samples that are not sufficiently optimized. It can be seen that, with the mask-purify process, the robust performance gradually improves.
> >
> > In addition, to further demonstrate the role of the mask phase, we plot the distribution of sample entropy in the mask and purify phases. Due to the difficulty of adding images in the comments, we have included the plotted images in the appendix. From Figure 6-7 in the appendix, it can be observed that after the mask phase, the entropy distribution of the samples is reduced compared to adversarial samples, alleviating adversarial characteristics. The samples no longer exhibit the phenomenon of high-confidence misclassification. Subsequently, in the purify phase, the entropy distribution moves closer to that of clean samples.
> >
> > Simultaneously, in the appendix, we have provided some visual results, as shown in Figure 8. After the mask stage, the confidence in misclassifying samples from the incorrect class decreases. In the attention maps, the focus on the incorrect class is disrupted, demonstrating the role of the mask stage in masking adversarial samples. Subsequently, in the purify stage, high-confidence predictions for the correct class gradually recover, and the attention maps progressively approach those of clean samples. Thus far, we believe the mask stage indeed initially plays a role in masking adversarial samples during the correction process, followed by the purify stage further enlightening the samples.

---

> ### Author Response · Authors · 2023-11-17
> **To Reviewer f1E9**
>
> **Q3: more experiments**
>
> **A3:**
> Due to time and resource constraints, in order to showcase results on additional datasets, we attempted experiments on the TinyImageNet-200 dataset. TinyImageNet-200 is designed for image classification tasks. It consists of 120,000 training images, 10,000 validation images, and 10,000 test images, with each image sized at 64x64 pixels. Results are shown in the tables below.
> We use **bold and italic** for the optimal value, **bold** for the suboptimal value.
>
> #TinyImageNet (resnet18)
>
> #TinyImageNet (resnet18)
>
> |            |      cln     |     FGSM     |     PGD      |      CW      |      DF      |
> |:----------:|:------------:|:------------:|:------------:|:------------:|:------------:|
> |    None    | **_51.01%_** |    2.00%     |    1.58%     |    0.00%     |    11.85%    |
> |  AT(FGSM)  |    29.00%    |  **13.45%**  |    12.29%    |    8.42%     |    18.19%    |
> |   AT(PGD)  |    28.49%    | **_13.70%_** |  **12.98%**  |    8.42%     |    17.96%    |
> | NO_defense |  **44.45%**  |    1.64%     |    0.00%     |    0.00%     |    12.34%    |
> |  SOAP(LC)  |    41.97%    |    7.97%     |    5.69%     | **_40.29%_** | **_40.65%_** |
> |  SOAP+ours |    30.90%    |    11.14%    | **_13.51%_** |  **39.65%**  |  **36.24%**  |
>
> ##TinyImageNet (widresnet28-10)
>
> |            |      cln     |     FGSM     |      PGD     |      CW      |      DF      |
> |:----------:|:------------:|:------------:|:------------:|:------------:|:------------:|
> |    None    | **_65.89%_** |     8.96%    |     0.26%    |     0.00%    |     9.66%    |
> |  AT(FGSM)  |    50.78%    |    24.00%    |    19.97%    |    10.25%    |    24.65%    |
> |   AT(PGD)  |    49.45%    | **_22.99%_** |    19.18%    |     8.95%    |    26.87%    |
> | NO_defense |  **64.32%**  |    15.70%    |     1.12%    |     0.00%    |     9.66%    |
> |  SOAP(LC)  |    47.23%    |    21.20%    |  **25.09%**  |  **44.12%**  | **_42.88%_** |
> |  SOAP+ours |    50.80%    |  **22.55%**  | **_29.45%_** | **_44.45%_** |  **42.20%**  |
>
> **Q4: The consistency of $L_{cls}$ and  $L_{ent}$ **
>
> **A4:** Thank you for your proposal. We check the position of the two losses in the article but no inconsistencies are found. In order to avoid misunderstanding, we will re-describe the two losses here.
>
> $L_{\text{cls}}(x, y)=-\sum_i y_i \log \left(p_i\right)$
>
> $L_{cls}$ refers to the cross-entropy loss for sample classification, calculating the correlation between the predicted values and the correct labels. We examine its occurrences in the text, and it consistently represents the cross-entropy loss for sample classification.
>
> $L_{\text{ent}}(x)=-\sum_i p_i \log \left(p_i\right)$
>
> $L_{ent}$ , on the other hand, refers to the predictive information entropy. Higher values indicate higher predictive entropy and lower confidence in predictions, and its calculation process does not require correct labels. In the text, when computing the weighting coefficients, we normalize the information entropy ${V}_{ent}$  by dividing it by log(N) (where N is the number of classes). This normalization operation might be overlooked, leading to potential misunderstandings.
>
> **Q5: Supplement about $\delta$**
>
> **A5:** We add $\delta$ content in the revision.
>
> **Q6: the effect of SNR and mutual information**
>
> **A6:** We consider the introduction of this problem to be highly meaningful. Regarding the exploration of entropy and error rate, corresponding experimental results are provided in [c]. Additionally, the calculation process of entropy is unsupervised, making it applicable in the test phase without labeled supervision. Therefore, we follow existing viewpoints and delved into the entropy properties of adversarial samples.
>
> Regarding the relationship between mutual information and error rate, mutual information is primarily used to measure the degree of mutual dependence between two random variables. The mutual information here can be seen as the correlation between features, and intuitively, correctly classified samples and misclassified samples may exhibit different properties. Currently, there is no exploration of the relationship between mutual information and error rate that I am aware of, and it is worth further investigation.
>
> Furthermore, regarding the relationship between signal-to-noise ratio (SNR) and error rate, in the field of communication systems, SNR does indeed have a certain correlation with error rates. However, when applied to the field of images, further exploration is still needed. Additionally, because the calculation process of image signal-to-noise ratio requires obtaining noise and the original image, when applied to the calculation process of adversarial samples, the corresponding participation of clean samples in supervision is needed. Therefore, its subsequent application may not be suitable for the test time stage, and careful consideration is required for its application.
>
> [c] https://arxiv.org/abs/2006.10726

---

### Author Response · Authors · 2023-11-20
**Reviewer Discussion Based on The Authors' Positive Feedback**

Dear Reviewers and AC,
Thanks so much for all your effort on reviewing our submission and giving us constructive comments. After reading the reviewers' comments, we think the reviewer's concerns are easily addressed and therefore we have carefully answered the reviewers' comments to eliminate your concerns.
If you still have questions after reading our response, please send us your further concerns, and we will make timely feedback in order to make everything as clear as possible before ending the discussion.

Thank you.
Authors

---

### Meta-Review · Area_Chair_d3KT · 2023-12-07

**Metareview:**

Summary: Inspired by the entropy properties of adversarial samples, the paper presents a max-min entropy strategy for test-time defense against adversarial samples.

**Justification For Why Not Higher Score:**

TBD

**Justification For Why Not Lower Score:**

TBD

---

### Decision · Program_Chairs · 2024-01-16

Reject